# Composite Coatings Applied to Fresh and Blanched Chayote (*Sechium edule*) and Modeling of the Drying Kinetics and Sorption Isotherms

**DOI:** 10.3390/foods13081178

**Published:** 2024-04-12

**Authors:** Yokiushirdhilgilmara Estrada-Girón, Angelina Martín del Campo-Campos, Emmanuel Gutiérrez-García, Víctor V. Fernández-Escamilla, Liliana Martínez-Chávez, Teresa J. Jaime-Ornelas

**Affiliations:** 1Departamento de Ingeniería Química, Centro Universitario de Ciencias Exactas e Ingenierías, Universidad de Guadalajara, Blvd. Marcelino García Barragán 1421, Col. Olímpica, Guadalajara 44430, Jalisco, Mexico; angelina.martindelcampo@academicos.udg.mx (A.M.d.C.-C.); emmanuel_gutierrez27@outlook.com (E.G.-G.); 2Departamento de Ciencias Tecnológicas, Centro Universitario de la Ciénega, Universidad de Guadalajara, Av. Universidad 1115, Col. Lindavista, Ocotlán 47820, Jalisco, Mexico; 3Departamento de Farmacobiología, Centro Universitario de Ciencias Exactas e Ingenierías, Universidad de Guadalajara, Blvd. Marcelino García Barragán 1421, Col. Olímpica, Guadalajara 44430, Jalisco, Mexico; liliana.mchavez@academicos.udg.mx; 4Departamento de Salud Pública, Centro Universitario de Ciencias Biológicas y Agropecuarias, Universidad de Guadalajara, Camino Ramón Padilla Sánchez 2100, Zapopan 45200, Jalisco, Mexico; dejesus.jaime@academicos.udg.mx

**Keywords:** *Sechium edule*, blanching, edible coatings, drying kinetics, sorption isotherms, modeling

## Abstract

Sustainable methods such as convective drying have regained interest in reducing the loss and waste of food produce. Combined with techniques like blanching and edible coatings, they could serve as useful tools in food processing development. Composite coatings comprising pectin, soy protein isolate, and xanthan gum were optimized using response surface methodology with the Box–Behnken design. This optimization aimed to investigate their effects on the moisture content, water activity, total color, and rehydration ratio of fresh and blanched chayote slices. Additionally, the study explored the modeling of the drying kinetics and sorption isotherms of chayote (*Sechium edule*) slices. Soy protein and xanthan gum were found to primarily influence the moisture content (ranging from 5.44% to 9.93%), and pectin influenced water activity (033 to 0.53) of the fresh-coated chayote, while pectin affected the a_w_ (2.13–8.28) and rehydration of the blanch-coated chayote. The optimized formulations for both fresh and blanched chayote were utilized to assess the drying kinetics behavior and sorption isotherms. The best fit (R^2^: 0.996 to 0.999) was achieved with the parabolic model for thin-layer materials. Furthermore, the sorption isotherms of chayote displayed a Type IV behavior, with the BET model being the most suitable for describing the sorption behavior of materials with low water activity. The predicted values offer valuable data for optimizing processing conditions to enhance the quality and stability of dried chayote.

## 1. Introduction

Chayote (*Sechium edule*), also known as chayote squash, is a vegetable that mainly contains water (95%), vitamins (C, B12, folate), minerals (potassium), dietary fiber, and some bioactive compounds of health benefit [1,2]. Globally, the main producers of chayote are Mexico, Costa Rica, Brazil, and the Dominican Republic, which predominantly export to Europe, the United States, and Canada [1].

Because of its high water content, chayote is a perishable vegetable usually preserved under refrigeration, which is not effective in the long term and has high energy costs. Moreover, drying is a conventional technology widely used to reduce food loss and wastage of perishable food produce, and lately, it has regained importance as a sustainable method to solve problems related to food preservation [2,3]. However, the direct exposure of products to hot air for long periods brings about undesirable food quality changes such as browning, shrinkage, and loss of nutrients, among others [2,3]. In this context, blanching used as a pretreatment of drying, besides its main purposes for enzyme inactivation and microbial load reduction, could improve texture, color, and pigment retention [2,4,5]. Investigations on a combined drying and edible coating, which is another effective technology for food preservation, mainly focus on the physicochemical properties of dehydrated products, but little is known about the drying mechanisms that govern the water mass transfer of coated organic materials. For instance, coated papaya slices with pectin, or kiwi with a combination of pectin and soy protein isolate, reduced the loss of vitamin C without affecting the mass transfer during drying. This resulted in products of improved quality [2,6,7].

On the other hand, the drying kinetics behavior of food, which is related to heat and mass transfer phenomena, is considered a useful tool to estimate drying conditions and prevent undesirable changes in food quality and stability during storage [8,9]. Many different mathematical models categorized as empirical, semi-theoretical, and theoretical equations, have been proposed to describe the drying behavior and to estimate the drying parameters of dried food [10]. The drying profiles of fruit and vegetables are best described by thin-layer models such as Page, Logarithmic, Henderson and Pabis, and Weibull, among others [2,11], whose simplicity or complexity depends on the number of constants. However, thin-layer models of an empirical or a semi-theoretical nature consider the external resistance to moisture transference between the solid and the hot air for a more precise prediction [2,11]. Furthermore, moisture content is an important factor that influences the shelf life of drying food during storage, as its exposure to variable relative humidity causes changes in quality and microbial safety. The equilibrium moisture content and water activity (a_w_) are related in a non-linear way to the moisture content by sorption isotherms, which helps to predict the shelf life of foods [3,12,13]. Sorption isotherms contribute to estimating the a_w_ and the monolayer moisture (M_0_), which are useful parameters to estimate the physical and chemical deterioration of dehydrated foods [9,14,15]. In this regard, two of the most used models applied to fit sorption isotherms for food applications are the BET (Brunauer, Emmett, and Teller) and GAB (Guggenheim, Anderson, and De Boer) equations [13]; the former is only linear in a limited range of a_w_ (<0.5), whereas the second fits more adequately to most food systems, with good accuracy and validity in a wider range from 0.1 to 0.9 [14]. 

Therefore, this research aimed to evaluate the effect of blanching and composite edible coatings as pretreatments of thin layers of chayote slices to enhance the effectiveness of drying and minimize changes caused by heat. The drying data were fitted with different mathematical equations to predict the stability of dried chayote slices. Additionally, sorption isotherms were built to determine the type of isotherms of coated chayote, and the BET, GAB, and Henderson models, recognized for fitting most food products within an aw range, were used to fit the sorption isotherms.

## 2. Materials and Methods

### 2.1. Materials and Reagents

The chayote fruits used in this work were purchased from a local market in the city of Guadalajara (Jalisco, Mexico). Citric pectin (P), xanthan gum (XG), and glycerol (G) were acquired from Meyer Chemical Reagents, Sigma-Aldrich, and Golden Bell, respectively. Soy protein isolate (SPI) was purchased from Food Technologies. All chemical salts—lithium chloride (LiCl), magnesium chloride (MgCl), potassium carbonate (K_2_CO_3_), sodium nitrite (Mg(NO_3_)_2_), potassium iodide (KI), and sodium chloride (NaCl)—were of reagent grade.

### 2.2. Chayote Preparation

Chayote fruits were washed and peeled with the help of a manual peeling device. Afterward, they were cut into thin slices (2 mm in thickness) and coated, either fresh or blanched, by immersion into the coating solution; uncoated fresh or blanched slices were considered as the treatment control. Blanching was performed following the procedure reported by Akonor and Tortoe [16] with some modifications. Briefly, the slices were placed on a stainless-steel mesh and immersed in distilled water at 85 ± 2 °C for 3 min; a probe tip thermometer was introduced to the blanching water to monitor the temperature. Afterward, the material was cooled down in a water bath at 10 °C for 1 min and drained in a stainless-steel mesh colander; the excess water was removed with absorbent paper.

### 2.3. Coating Solution and Application

Coating solutions were prepared with pectin, soy protein isolate, and xanthan gum (Table 1), adding glycerol as a plasticizer and distilled water to adjust to 100% (*w*/*v*). Before its use, SPI was dissolved in water at 85 °C for 30 min and cooled down to 25 °C [2,17]. The mixture of biopolymers was homogenized under constant agitation on a stirring plate (OS20-S, DLAB Scientific Inc., Riverside CA, USA) until their complete dissolution. The chayote slices were immersed in the coating solutions for 1 min and placed on a stainless-steel grid to remove the excess solution. The coated slices were dehydrated in a convective dryer (DESH304/032014, TODUMEX, Guadalajara, MEX) under a hot-air stream at 65 °C and a flowing rate of 0.7 m/sec until a constant weight was reached. After drying, all samples were stored inside metalized bags until further analysis.

### 2.4. Measurements of Moisture, a_w_, Color, and Rehydration Capacity

The moisture content of the samples was determined by the gravimetric method following the procedure of Roudaut [18]. An Aqualab equipment (Decagon Devices, Pullman, Whitman, WA, USA) was used to measure the water activity (a_w_) of samples [19]; all measurements were performed in triplicate.

The color of powder chayote was measured with a colorimeter (Konica Minolta, CR410, Tokyo, Japan) using the CIELab color space (*L**, *a**, and *b**); readings were taken at five random points [20]. Total color change (∆E) was calculated as $∆E=\sqrt{{∆a}^{*2}+{∆b}^{*2}+{∆L}^{2}}$ [21], where L* is the luminosity [black: *L** = 0 to white: *L** = 100) on a black–white scale. The color coordinate *a** took positive values for red and negative for green colors while *b** was positive for yellow and negative for blue in the color space [20].

The rehydration capacity was measured according to Roudaut [18]. A total of 2 g of dried chayote was added to distilled water at room temperature for 5 h. Later, samples were removed, and the excess water was cleaned with absorbent paper. The rehydration ratio based on water gained with time was calculated as follows: $Rehydration\;ratio=\frac{W_{i}}{W_{d}}$, where $W_{d}$ is the final weight of the hydrated sample (g) and $W_{i}$ is the initial weight of the dried sample (g) [22].

### 2.5. Experimental Design and Optimization

The coating formulation applied to chayote slices was optimized by the response surface methodology using a Box–Behnken design with three independent variables: pectin (P), soy protein isolate (SPI), and xanthan gum (XG), which involved 15 runs with three center points (Table 1) in two blocks; the compounds’ percentages were selected after preliminary tests. The dependent variables were the moisture content, water activity, total color, and rehydration capacity of the fresh and blanched chayote slices. The optimized formulation was prepared similarly as mentioned above to conduct the modeling of drying and sorption isotherms. 

The following quadratic polynomial model was used to study the regression model:$$\begin{array}{cc} Y= & \beta_{o\;}+\beta_{1\;}X_{1}+\beta_{2\;}X_{2}+\beta_{3\;}X_{3}+\beta_{11\;}{X_{1}}^{2}+\beta_{22\;}{X_{2}}^{2}\beta_{11\;}{X_{1}}^{2}+\beta_{33\;}{X_{3}}^{2}+\beta_{12\;}X_{1}X_{2}+\\ & \beta_{13}X_{1}X_{3}+\beta_{23\;}X_{2}X_{3} \end{array}$$
where *Y* = dependent variable; *β*_o_, is the intercept; *β*_1_, *β*_2_, and *β*_3_ are the linear coefficients; *β*_12_, *β*_22_, *β*_23_ are interaction coefficients; *β*_11_, *β*_22_, *β*_33_ are the quadratic coefficients; *X*_1_, *X*_2_, and *X*_3_ are the factors indicated in Table 1 as P (0–2.0% *w*/*w*), SPI (0–2.0% *w*/*w*), and XG (0–0.3% *w*/*w*), respectively. All measurements of the dependent variables were performed in triplicate, and the results are reported as mean values ± standard deviation.

### 2.6. Drying Kinetics and Modeling 

The drying curves were constructed by measuring the weight loss of samples during the drying period. Data were collected every 30 min during the first 2 h and then every hour until reaching a constant weight. The drying rate of the chayote slices was calculated as follows [8]:(1)$$DR=\frac{M_{t}-M_{t+∆t}}{∆t}$$
where *DR* is the drying rate in $\frac{kg\;of\;evaporated\;water}{kg\;of\;dry\;matter\;*\;min}$, *M_t_* is the moisture content on a dry basis (*d.b.*) at any time, $M_{t+∆t}$ is the moisture content on *d.b.* at time t+∆t, *t* is the time in minutes, and Δ*t* is the increment time. Through a mass balance, *M_t_* was calculated as follows [23]:(2)$$M_{t}=\frac{\left(W_{i}M_{i}\right)-W_{evp}}{(1-M_{i})W_{i}}$$
where *W_i_* is the initial weight of the chayote slices in g; *M_i_* is the initial moisture content in the wet base (*w.b*.); *W_evp_* is the mass of the evaporated water, which is the result of *W_i_* − *W_t_*, where *W_t_* is the weight of the chayote slices at a time *t*. These data were used to create graphs of moisture versus time and the drying rate versus moisture. 

The experimental data were fitted to four models reported elsewhere [10]: (3)MR=aexp⁡(−kt)
(4)$$MR=exp\left(-{\left(\frac{1}{b}\right)}^{a}\right)$$
(5)$$MR=\exp\left({-kt}^{n}\right)$$
(6)$$MR=a+bt+{ct}^{2}$$

Equations (3)–(6) are the Henderson and Pabis, Weibull, Page, and parabolic models, respectively, where *a*, *k*, *b*, *n*, and *c* are constants. The fitting of the equations was performed with the software OriginPro version 2018b (OriginLab Corp., Northampton, MA, USA). The best model describing the drying kinetics of chayote slices was that exhibiting the highest correlation coefficient (R^2^).

The moisture ratio (MR) is defined as follows:(7)$$MR=\frac{M_{t}-M_{e}}{M_{i}-M_{e}}$$
where the subscripts *t*, *e*, and *i* represent moisture at time *t*, moisture in equilibrium, and initial moisture, respectively (on dry basis). *M_e_* is the last moisture value obtained from the drying kinetics and represents the humidity in equilibrium with the drying air.

### 2.7. Determination of the Diffusion Coefficient

The diffusion coefficient of water toward the drying air during the drying time was calculated as follows [24]:(8)$$D=\frac{K_{t}L^{2}}{\pi^{2}}$$
where *D* is the diffusion coefficient in mm^2^ h^−1^, *K_t_* is the drying coefficient at time *t* in h^−1^, and *L* is the thickness of the slices in mm. The drying coefficient was determined as follows: (9)$$K=-\frac{\ln\left(\frac{M_{t}-M_{e}}{M_{i}-M_{e}}\right)}{t}=-\frac{\ln MR}{t}$$

### 2.8. Sorption Isotherms 

Sorption isotherms were constructed based on the gravimetric method. Samples of the dehydrated chayote slices were placed inside desiccators with saturated solutions to maintain the specific relative humidity at 25 °C. The water activity range was set from 0.113 to 0.75 using different salt solutions: LiCl = 0.113, MgCl = 0.328, K_2_CO_3_ = 0.432, Mg(NO_3_)_2_ = 0.529, KI = 0.689, and NaCl = 0.75 [18]. The relative humidity for each salt solution at 25 °C was around 11.1%, 32.8%, 44%, 55.8%, 68%, and 75.5%, respectively [25,26]. Samples were weighed periodically until reaching a constant weight, determined when the difference between two consecutive measurements was less than 0.5%. The experimental data were fitted using three models: BET (10), GAB (11), and Henderson (12), all of them applied to food systems:(10)$$M=\frac{M_{0}Ca_{w}K}{\left(1-Ka_{w})(1-{Ka}_{w}+C{Ka}_{w}\right)}$$
(11)$$M=\frac{M_{0}Ca_{w}}{\left(1-a_{w})(1-Ca_{w}-a_{w}\right)}$$
(12)$$\log\left[\log\left(\frac{1}{1-a_{w}}\right)\right]=\log C_{H}+b\log\left(100M\right)$$
where *M* is the moisture content in *d.b.* (%), *a_w_* is the water activity, *M*_0_ is the monolayer moisture content, and *C*, *K*, *C_H_*, and *b* are the constants of the models. The values of the coefficients were calculated from the linearized equations according to Silva et al. [27].

The fitting of the models was evaluated with the root mean value of the squared error (*RSME*), which was calculated with the following expression:(13)$$RSME=\sqrt{\left[\frac{1}{n}\sum_{i=1}^{n}{\left(M_{pred,\;i}-M_{exp,i}\right)}^{2}\right]}$$
where *M_pred_* is the moisture value estimated by the model, *M_exp_* is the experimental moisture value (observed), and h is the number of the experimental data [28]. Thus, the best fitting is achieved in those models with the lowest *RSME* value [29].

### 2.9. Statistical Analysis 

The experimental data were statistically analyzed with the analysis of variance ANOVA using the Statgraphics Centurion XV software, version 19 (Statpoint Technologies, Warrenton, VA, USA). Differences between means were calculated with the Fisher’s Least Significant Difference (LSD) test (*p* < 0.05).

## 3. Results and Discussion

### 3.1. Physicochemical Measurements and Rehydration Capacity

The moisture content ranged between 5.44 and 9.93 and between 6.20 and 9.31% for fresh-coated and blanch-coated slices, respectively (Table 2). Particularly, the formulations (treatments 7, 8, and 10) with a high percentage of xanthan gum exhibited a major effect on this parameter, whereas the moistures of the uncoated (control) fresh or blanched slices had low percentages (5.56 and 6.24%, respectively) comparable to those reported for dehydrated chayote [18]. The a_w_ of the fresh-coated slices ranged between 0.33 and 0.53, and some treatments were closer to the limiting a_w_ value (<0.6) indicated for food safety; for the blanch-coated slices, a_w_ was much lower, ranging between 0.24 and 0.37; this is because blanching improves the drying process by increasing permeability in plant vegetative tissue and, consequently, increases the rate of water removal [5,9].

After drying, the natural green color of the chayote diminished for the fresh uncoated slices, while blanching improved and maintained the natural pigments (Figure 1), as blanching inactivates enzymes that modify or cause color changes [19]. With the application of the coating, various treatments of the fresh-coated slices showed ΔE (total color difference) values above 3.0 (Table 2). Treatments with a high percentage of soy protein isolate had superior values (treatment 4: ΔE = 8.28), as protein imparts an ochre color to the film-forming solutions, and after drying, samples acquire a yellowish tone attributed to Maillard reactions [12,15]. According to Pathare et al. [20], if ΔE is greater than 3.0, the total color difference between a sample and the standard is perceptible at sight. Moreover, blanch-coated slices showed ΔE values of 3.92–7.96, suggesting the effect of both blanching and soy protein isolate in formulations containing protein.

The rehydration ratio (RR) of the fresh-coated slices was between 5.1 and 7.5 (Figure 2), which was higher (treatments 3 and 8) compared to that of the fresh uncoated slices. On the contrary, blanching resulted in a lower RR between 3.5 and 5.7, but various treatments behaved more like the blanched slices. The decrease in RR could be associated with the fact that the coating occupied rehydration sites and, therefore, the ratio was smaller. For dehydrated chayote, Akonor and Tortoe [16] reported an RR of ~4.8, and by adding a coating, the RR could be improved as demonstrated here. It is known that the rehydration process itself is a complex phenomenon, and considering that the coating offers additional resistance, the process becomes even more complex. 

### 3.2. Optimized Coating

For the fresh-coated chayote, the significant factors (*p* < 0.05) for moisture were protein and the pectin–gum and protein–gum interactions (Table 3); therefore, the lowest values of moisture were achieved at low percentages of xanthan gum and a high protein content. The significant effects for a_w_ were pectin and the quadratic effect of pectin–pectin and protein–gum interactions (*p* < 0.05). This is attributed to pectin, which is a hygroscopic compound, and due to its chemical structure, it has hydrophilic groups that absorb greater amounts of water from the environment [9,12,18]. For ΔE, the significant effects were pectin, xanthan gum, and pectin–pectin. This effect is associated with the longer drying times of the fresh slices to achieve constant weight, as more time is needed to dry the hydrocolloid layer, and, consequently, there is greater color, particularly with formulations of high protein. The surface response plots of the parameters with significant factors are shown in Figure 3. Although there was variability in the rehydration capacity, the statistics did not show the significance of the factors or their combinations. The surface response plots of the parameters with significant factors are shown in Figure 3a–c, and as observed, the concave curvature is more pronounced for moisture and a_w_ than that of color, meaning a better fit of the quadratic model for the former.

In contrast, for the blanch-coated slices, the only significant factor for a_w_ and rehydration was pectin, indicating that this compound exerts the main effect on these parameters; moreover, the surface plots exhibited less curvature (Figure 4a,b) compared to the fresh-coated samples, suggesting that blanching confers greater stability to the vegetative tissue in terms of color and, therefore, minimizes physicochemical changes during drying.

The regression coefficients, sum of squares, *p*-values, and fitting of the model of the significant factors for each parameter are summarized in Appendix A. The equations for each parameter are listed below.

Fresh-coated slices:Moisture = 6.36917 − 0.615833*X_1_* + 4.04917*X_2_* − 12.2056*X_3_* − 0.138333 *X_1_*^2^ + 0.37*X_1_X_2_* + 6.7*X_1_X_3_* − 2.37833*X_2_*^2^ + 6.86667*X_2_X_3_* + 22.0741*X_3_*^2^
a*_w_* = 0.442917 + 0.102917*X_1_* + 0.0241667*X_2_* − 0.297222*X_3_* − 0.0595833X_1_^2^ + 0.015*X_1_X_2_* + 0.216667*X_1_X_3_* − 0.0470833*X_2_*^2^ + 0.433333*X_2_X_3_* − 1.09259*X_3_*^2^
Color = 3.92583 + 0.857083*X_1_* − 0.0541667*X_2_* + 4.93056*X_3_* − 0.759167*X_1_*^2^ + 0.8625*X_1_X_2_* + 6.06667*X_1_X_3_* − 0.311667*X_2_*^2^ + 3.28333*X_2_X_3_* − 43.074*X_3_*^2^
Rehydration = 7.2225 − 0.7325*X_1_* − 0.16125*X_2_* − 7.29167*X_3_* − 0.035*X_1_*^2^ + 0.1225*X_1_X_2_* + 1.18333*X_1_X_3_* − 0.1125*X_2_*^2^ + 3.16667*X_2_X_3_* + 10.5556*X_3_*^2^

Blanch-coated slices:Moisture (%) = 5.14083 + 0.917083*X_1_* + 1.63083*X_2_* + 15.2139*X_3_* + 0.278333*X_1_*^2^ − 0.4225*X_1_X_2_* − 3.73333*X_1_X_3_* − 0.544167*X_2_*^2^ + 0.45*X_2_X_3_* − 38.5185*X_3_*^2^
a_w_ = 0.289274 + 0.0426302*X_1_* + 0.0498552*X_2_* − 0.0770694*X_3_* − 0.0190385*X_1_*^2^ + 0.0135187*X_1_X_2_* + 0.0645833*X_1_X_3_* − 0.0269385*X_2_*^2^ + 0.004*X_2_X_3_* − 0.244213*X_3_*^2^
Color = 6.39792 + 0.402917*X_1_* − 0.542083*X_2_* − 12.1889*X_3_* + 0.420417*X_1_*^2^ − 0.4225*X_1_X_2_* − 1.9*X_1_X_3_* − 0.109583*X_2_*^2^ + 2.93333*X_2_X_3_* + 29.9074*X_3_*^2^
Rehydration = 4.38708 + 0.438333*X_1_* − 0.749167*X_2_* − 4.12778*X_3_* + 0.0395833*X_1_*^2^ − 0.1925*X_1_X_2_* + 2.28333*X_1_X_3_* + 0.354583*X_2_*^2^ + 4.51667*X_2_X_3_* − 7.35185*X_3_*^2^

By a multi-response analysis, the optimized formulation based on the greatest desirability for the significant effects provided a combination based on soy protein isolate and xanthan gum with 0.82 of desirability (Figure 3d) for the fresh-coated slices and a formulation consisting of pectin, soy protein isolate, and xanthan gum with 0.71 of desirability (Figure 4c) for the blanch-coated slices These formulations were used as coatings of chayote slices to model the drying kinetics and sorption isotherms. 

### 3.3. Drying Kinetics 

For the moisture ratio (MR), the fresh-coated slices exhibited a superior average MR of 0.230 g water/g of sample (*d.b*.) after 300 min of drying (Figure 5a); this value was almost twice that displayed by the fresh uncoated slices (0.132 g water/g of sample, *p* < 0.05) that reached a constant weight within a shorter period. Under the convective drying of fruits and vegetables, the removal of water is associated with the great amounts of water and solids in the superficial layers of the vegetative tissue that modify vapor pressure and create different fluxes of evaporated water [24,30]. However, the film layer formed on the surface of the intact plant tissue of the coated slices offered additional resistance and acted as a barrier against mass transfer. Similar results were reported for papaya or kiwi fruit coated with biopolymers, indicating that the coating offers extra resistance to heat transfer and water removal [4,6,7].

Between fresh and blanched uncoated slices (control treatments), the MR of the latest was only slightly above that of the former; this behavior may be attributed to blanching that affects plant tissue, which loses its turgor, and, therefore, some sites are no longer available for mass transfer because of the collapse of pores and the shrinkage during drying [24,28,29]. For materials like aonla shreds, blanching reduces the drying time because of the disruption of cell walls that leads to an increase in the water diffusion rate [31]. Moreover, the blanch-coated slices had a significant (*p* < 0.05) decrease in MR with 0.0069 g water/g of sample and reached a constant moisture after 310 min of drying. This suggests that the film layer supported the debilitated vegetative tissue after blanching and prevented its collapse, thus, allowing for a faster removal of water molecules that evaporated more easily. In addition to heat treatment, it causes the breakdown of the hydrogen bonding of water molecules within food, contributing to the evaporation process, as reported for biopolymers such as starch [27].

Although no data have been reported previously on the use of edible coatings as a pretreatment for chayote dehydration, for fresh slices or pieces of chayote either treated with brine solutions or by osmotic dehydration, longer drying times (>300 min) are needed to reach a constant weight [12,18,29]. With other food materials such as blanched pumpkin slices [30], the drying times were like those reported here; nonetheless, the drying time varies depending on the drying conditions, geometry, and thickness of the samples. Regarding the thickness of the coatings, the optimized solutions yielded films of 0.083 and 0.076 mm for the fresh and blanched slices, respectively. Although the coatings protected the chayote slices from unwanted damage during the air-drying process, increasing the thickness of the coating could lead to an increase in the drying times.

From the drying rate curves (Figure 5b), we can observe two main stages of drying: (1) a drying period where the rate of water removal is constant from 18 to 5.5 kg water/kg of dry matter (within the first 300 min), which is associated with the free water; and (2) a falling rate period that represents the monolayer moisture, which evaporates to a certain level (the final stage of drying) [32]. During the constant-rate period, the drying rates of the fresh-coated and both controls behaved alike and within a close range from 0.056 to 0.040 kg water/kg of dry matter*min, which corresponded to 16.09 and 4.95 kg water/kg of dry matter*min, respectively; a similar pattern was also observed during the falling rate period. Conversely, the drying rate of the blanch-coated slices did not present a constant period but exhibited a sharp decrease from 0.092 to 0.003 kg water/kg of dry matter*min; meaning that water was removed more easily, as previously attributed to the film layer. Furthermore, the critical moisture (Xc) is an important parameter that can be obtained from the drying rate curves and represents the value at which the constant rate changes to the falling rate. During drying, there is a variation in its mechanism as a function of Xc: before the material reaches Xc, the water that evaporates is that found on the surface of the food; after Xc, the water that evaporates is that contained inside the food, and it is transferred through the pores of the material from inside out [24,27]. In this regard, all samples exhibited similar Xc values (5.0–5.6 kg water/kg of dry matter), except the blanch-coated slices that achieved the highest Xc with 9.2 kg water/kg of dry matter (Table 3). This behavior indicates that the surface film accelerates the mass transfer phenomenon, thus moving more quickly to the falling rate period, which is usually the longest period of the curve and governs the drying process. Because of their ability to delay gas transmission, polysaccharide-based coatings may help to promote the removal of water during drying [4,24,32]. Further, protein-based coatings provide functional groups that form interactions depending on the amino acids they contain; thus, offering a barrier to oxygen to a greater or lesser extent [16,19,29] and soy protein isolate act as an excellent gas barrier that provides superior properties to edible films [17].

### 3.4. Modeling of the Drying Process

Different mathematical models have been proposed to predict the behavior of food during the drying process and to design or improve drying systems, as reported elsewhere [10,30]. Figure 6a,b depict plots of MR versus time with the fitting of models for all coated and uncoated chayote slices; the values of the drying model parameters and constants for all equations are presented in Table 4. The predictions of Equations (4)–(6) reproduce the experimental data quite well within the time range studied, whereas deviations from Equations (4) and (5) are observed at short times in the inserts of the figures (Figure 6a,b); Equations (5) and (6) fit the data well in all the time ranges studied. Moreover, deviations from Equation (3) (Henderson and Pabis model) are observed over all the time ranges studied. Although, for both treatments (coated) and their controls, all the tested models yielded values of R^2^ close to the unit (Table 4), the best fit was achieved with Equation (6) (the parabolic model), which yielded R^2^ values in the range of 0.996–0.999; as this model is empirical, its parameters have no physical meaning, as well as the Weibull (Equation (4)) and Page (Equation (4)) models. 

On the other hand, the Henderson and Pabis model (Equation (3) has a physical meaning, where the value of k is related to the effective diffusivity in porous media [23]. However, the deviations from the Henderson and Pabis model and the experimental data are evident. The departure from the Henderson and Pabis model could be related to the type of Fickian diffusion mechanism of water through the porous pores of chayote slices. Moreover, notice that the k value of the blanch-coated samples is greater than those of other treatments (Table 4); this indicates that drying occurs more rapidly as a consequence of the increase in the effective diffusivity [10,23]. Thus, it becomes evident that a better prediction of drying kinetics requires more fitting parameters, where the empirical models can be used to predict the drying times and the final moisture of the chayote. These models were selected because they are reported as having an excellent correlation between experiments and models for the drying kinetics of foods [15]. The results show a very good fit with quadratic expressions. This indicates that all the drying curves for these systems can be represented approximately by a parabolic semi-diffusion model, which is expressed by analogy with the quadratic diffusive term of the Law of Fick.

### 3.5. Diffusion Coefficient

The diffusion coefficient versus time of both fresh-coated and blanch-coated chayote slices and their treatment control is reported in Figure 7. As can be noticed, the diffusion coefficient of the blanch-coated slices shows an exponential increase from the beginning and reaches its maximum value after 5 h of drying; therefore, the solute moves more easily in the food matrix, increasing the rate of water removal. In comparison, the application of the coating on fresh slices delayed the migration of water, causing a slight increase in the diffusion coefficient after 8 h of drying. The diffusion coefficients for blanch-coated chayote were higher than those of the other treatments; this may be associated with the blanching process that influenced the moisture content of chayote slices and promoted the mass transfer through the hydrophilic nature of the coating molecules; this is consistent with other food systems like potatoes [33]. The drying behavior in the falling rate period also confirms that the moisture transfer (rate of drying) is controlled by diffusion mechanisms. These results agree with those conducted by Falade and Solademi [33], who suggested that the drying process of most food materials occurs predominately in a falling rate period [9]. 

In the same context, water diffusion coefficients were strongly affected by the coating; therefore, the diffusion is dominated by the physical mechanism governing moisture movement in the drying process. Moreover, the application of a surface coating has been reported to increase the resistance of fruit skin to gas diffusion and the creation of a modified internal atmosphere. These results agree with observations of earlier researchers on the drying of other vegetables [8]. Hence, the knowledge of effective moisture diffusivity is necessary for designing and modeling mass-transfer processes such as the dehydration, adsorption, and desorption of moisture during storage [24]. The drying rate is variable during drying due to the different moisture transport mechanisms such as surface diffusion [24]. 

### 3.6. Sorption Isotherms and Modeling 

Figure 8 shows the equilibrium moisture versus water activity for both fresh-coated and blanch-coated slices and their treatment control. Based on their shape, the adsorption isotherms are classified into five types, but the most common for foods are types II and IV [34]. As observed, the isotherms of chayote slices exhibit shapes like those of type IV, which describes the adsorption by an expandable hydrophilic solid until reaching a maximum of hydration sites [34], whereas most foods show type-II isotherms.

The sorption isotherms of dried chayote slices are crucial to determine the stability of the product during the shelf life. The isotherms of the fresh-coated samples and both controls were quite similar, since at a_w_ of 0.53, they achieved an equilibrium moisture of around 27.9–28.5%. In contrast, the blanch-coated slices had much lower equilibrium moisture (22.4%) at the same value of a_w_; this shows how the water is adsorbed in the food material in the fresh intact tissue and after its exposure to heat during blanching. From the calculated RSME, in general, the BET model yielded lower values, indicating that this model more accurately predicts the behavior of dehydrated foods in a narrow a_w_ range (0.05–0.55), and among treatments, the blanch-coated slices showed the lowest value of 1.11 (Table 5). Moreover, the GAB and Henderson models, which are more useful in a wider range (a_w_: 0.05–0.9) [14], yielded a higher RSME; therefore, there was a higher level of dispersion of the residuals. Additionally, the predicted M_0_ or monolayer moisture was minimal in the blanch-coated slices with the BET model, suggesting the lower availability of active sites for binding with water. This value is a critical parameter because it is related to moisture where food deterioration reactions begin to occur.

In this context, the parameter *C* of the GAB equation represents the absorption constant related to the energy of interaction between the first and additional sorbed water molecules; thus, the larger the value of *C,* the stronger the interactions between the water molecules of the monolayer and the binding sites. As the fresh-coated and blanched slices exhibited the highest values, this suggests stronger interactions and more polar sites, indicating that the water is more strongly adsorbed and opposes a greater resistance during drying [18]. In contrast, the blanch-coated slices had lower *C* values, which is because the evaporation occurred more quickly, facilitating their dehydration. Otherwise, the parameter K is a correction factor for the multilayer molecules relative to the liquid; when *K* = 1, the molecules beyond the monolayer have the same characteristics as the pure liquid [13]. For all treatments, *K* approximates to one meaning: there are small differences between the molecules in the multilayer and unbounded molecules in the bulk liquid. Hence, both parameters *C* and *K* give information on the efficacy of the fitting of the GAB model. In addition, the Henderson model satisfactorily fitted the experimental data, with RSME values <10%, and could be useful to describe the behavior of coated and dried food produce.

## 4. Conclusions

To extend the shelf life of chayote, fresh and blanched slices were coated with composite edible film-forming solutions prior to convective drying. The coating components influenced the measured parameters: soy protein isolate and xanthan gum mainly increase moisture and water activity, while pectin and soy protein isolate may enhance or impart color, depending on whether they are applied to fresh or blanched products. Therefore, optimized coating formulations are needed because formulations intended for fresh produce may not be suitable after blanching foods. The coating formed on the surface of the vegetative tissue acted as a support for the food matrix and favored the faster removal of water for a more efficient mass transfer mechanism. Of the four drying models applied, the parabolic model yielded the best fit, with higher R^2^ values, for coated fresh and blanched slices. The sorption isotherms exhibited a sigmoidal type-IV shape and were better fitted to the BET model that predicted the lowest water activity and monolayer values. This investigation constitutes an approach to the use of combined technologies suitable to reduce the loss and waste of food produce.

## Figures and Tables

**Figure 1 foods-13-01178-f001:**
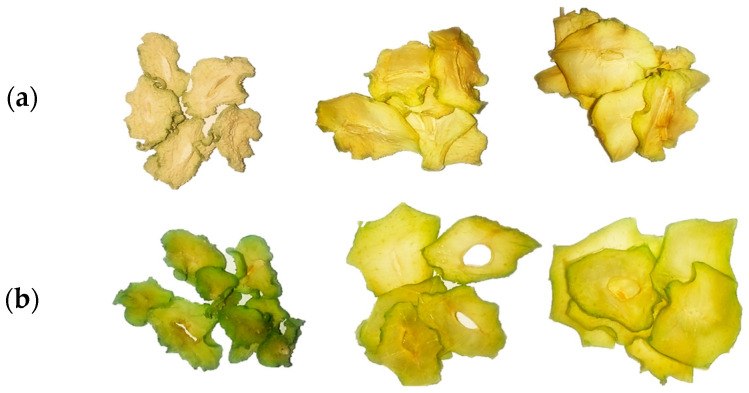
Physical appearance of dried chayote slices from left to right. Fresh: (**a**) uncoated, treatment 8, and treatment 11; blanched: (**b**) uncoated, treatment 8, and treatment 11.

**Figure 2 foods-13-01178-f002:**
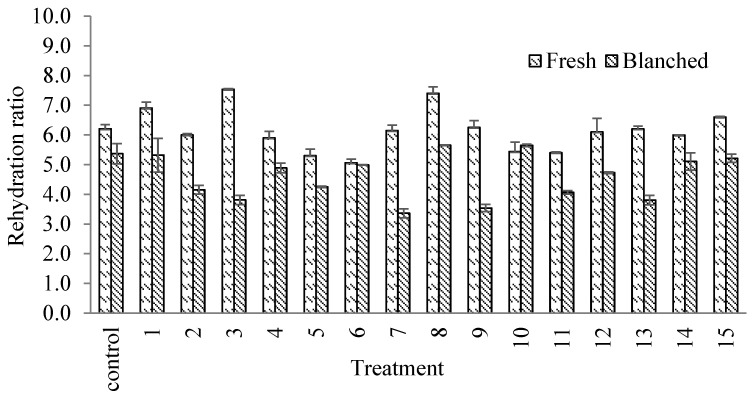
Rehydration ratio of uncoated (control) and coated fresh or blanched chayote slices.

**Figure 3 foods-13-01178-f003:**
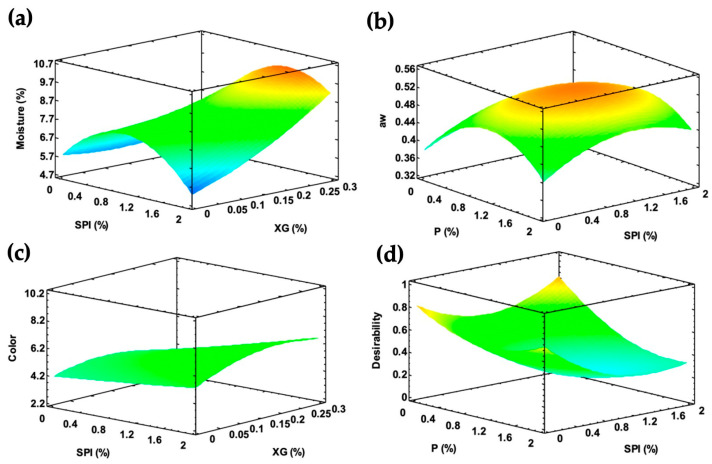
Surface response plots for (**a**) moisture content, (**b**) water activity, (**c**) total color, and (**d**) desirability of the optimized formulation of fresh-coated chayote slices.

**Figure 4 foods-13-01178-f004:**
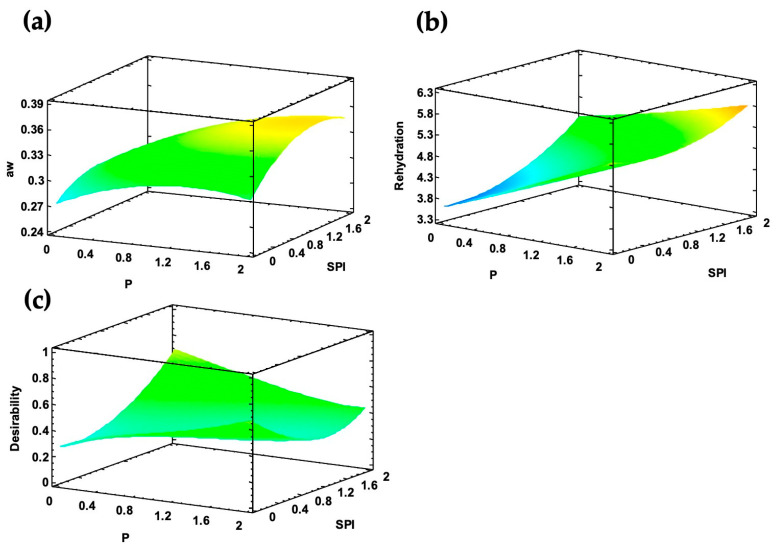
Surface response plots for (**a**) water activity, (**b**) rehydration, and (**c**) desirability of the optimized formulation of blanch-coated chayote slices.

**Figure 5 foods-13-01178-f005:**
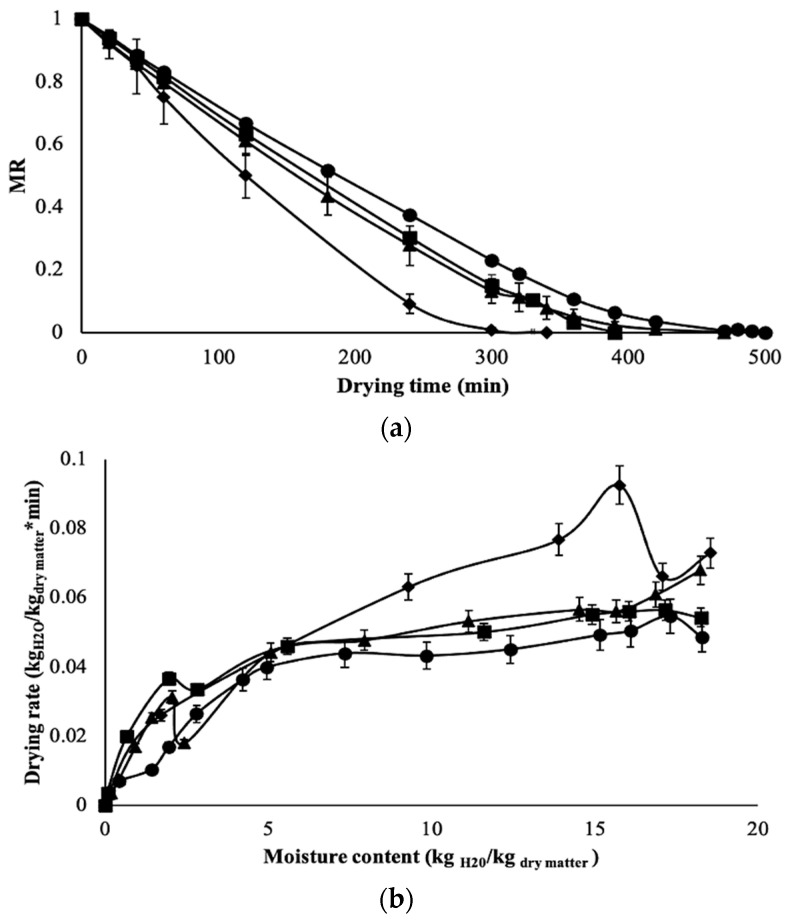
(**a**) MR and (**b**) drying rate behavior of fresh (▲), fresh-coated (●), blanched (■), and blanch-coated (♦) chayote slices; points are averages in the figure. The solid lines are an aid to the eye.

**Figure 6 foods-13-01178-f006:**
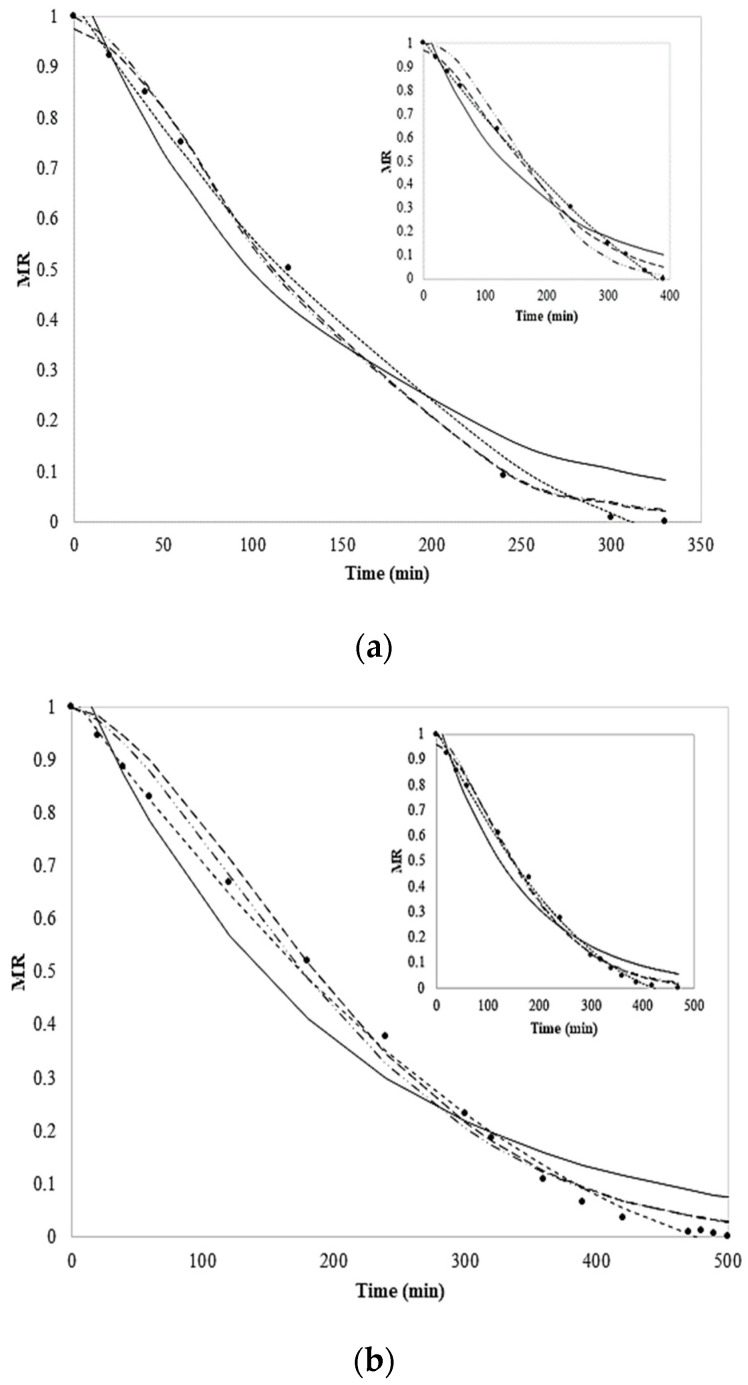
Modeling of the drying kinetics for (**a**) fresh-coated and (**b**) blanch-coated chayote slices. The lines represent the fitting of the models: (---) Parabolic, (—) Henderson and Pabis, (— –) Page, and (—∙∙) Weibull. The insert shows the drying kinetics of fresh dried chayote slices.

**Figure 7 foods-13-01178-f007:**
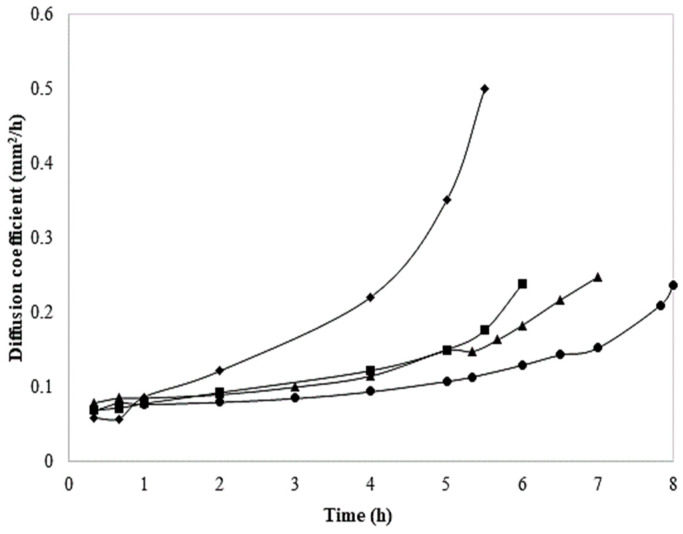
Diffusion coefficient of (▲) fresh, (●) fresh-coated, (■) blanched, and (♦) blanch-coated chayote slices. The solid lines are an aid to the eye.

**Figure 8 foods-13-01178-f008:**
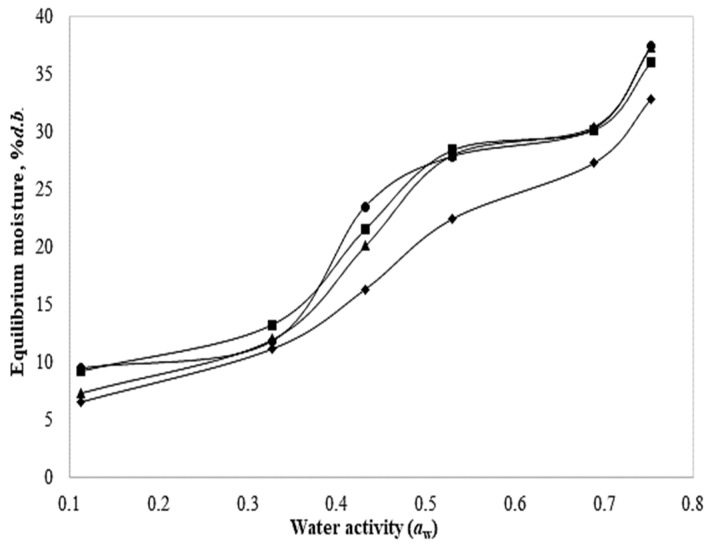
Sorption isotherms of (▲) fresh, (●) fresh-coated, (■) blanched, and (♦) blanch-coated chayote slices. The solid lines are an aid to the eye.

**Table 1 foods-13-01178-t001:** Box–Behnken design for the preparation of composite edible coatings.

Run	Coating Components (%)		
	P (X_1_)	SPI (X_2_)	XG (X_3_)
1 *	1	1	0.15
2 *	1	1	0.15
3	0	1	0.00
4	2	2	0.15
5	1	2	0.00
6	2	0	0.15
7	0	1	0.30
8	1	2	0.30
9	0	0	0.15
10	2	1	0.30
11 *	1	1	0.15
12	2	1	0.00
13	1	0	0.30
14	1	0	0.00
15	0	2	0.15

* Center points of the design.

**Table 2 foods-13-01178-t002:** Physicochemical parameters of fresh and blanched chayote slices after coating and dehydration by convective drying.

Treatment	Moisture (%, *d.b*.)		a_w_		ΔE	
	Fresh	Blanched	Fresh	Blanched	Fresh	Blanched
Uncoated	5.56 ± 0.33	6.24 ± 0.4	0.36 ± 0.01	0.35 ± 0.02	-	-
1	8.07 ± 0.53	8.84 ± 0.65	0.47 ± 0.06	0.33 ± 0.01	6.20 ± 0.66	5.32 ± 0.87
2	8.94 ± 0.17	7.79 ± 0.03	0.53 ± 0.03	0.36 ± 0.01	7.34 ± 1.31	4.16 ± 0.56
3	7.51 ± 0.85	7.58 ± 0.34	0.45 ± 0.06	0.34 ± 0.04	5.11 ± 0.71	5.82 ± 0.99
4	6.68 ± 0.44	9.26 ± 0.01	0.46 ± 0.05	0.30 ± 0.03	8.28 ± 0.28	4.63 ± 0.26
5	6.45 ± 0.54	6.20 ± 0.50	0.38 ± 0.00	0.30 ± 0.03	4.04 ± 0.31	5.46 ± 0.07
6	6.24 ± 0.42	9.17 ± 1.20	0.44 ± 0.05	0.30 ± 0.01	5.39 ± 0.97	7.84 ± 0.88
7	9.23 ± 0.48	7.85 ± 0.72	0.34 ± 0.01	0.24 ± 0.01	2.80 ± 0.15	6.07 ± 0.45
8	9.57 ± 0.99	7.54 ± 0.24	0.53 ± 0.06	0.34 ± 0.02	6.33 ± 0.45	5.35 ± 0.39
9	5.74 ± 0.10	6.38 ± 0.29	0.36 ± 0.02	0.27 ± 0.01	2.69 ± 0.06	5.44 ± 0.92
10	9.93 ± 0.56	7.04 ± 0.08	0.40 ± 0.01	0.29 ± 0.01	4.64 ± 0.49	6.09 ± 0.49
11	8.96 ± 0.73	7.21 ± 0.10	0.52 ± 0.04	0.34 ± 0.02	5.54 ± 0.23	7.96 ± 0.27
12	6.19 ± 0.06	9.31 ± 0.74	0.38 ± 0.01	0.37 ± 0.03	3.31 ± 0.02	6.98 ± 0.12
13	6.51 ± 0.14	7.58 ± 0.59	0.36 ± 0.03	0.32 ± 0.01	3.80 ± 0.22	5.08 ± 0.66
14	5.44 ± 0.37	6.55 ± 0.27	0.47 ± 0.01	0.28 ± 0.01	3.48 ± 0.72	6.95 ± 0.56
15	6.69 ± 0.86	8.43 ± 0.97	0.33 ± 0.02	0.26 ± 0.02	2.13 ± 0.04	3.92 ± 0.25

Moisture (%, d.b.): Percentage moisture in dry basis; a_w_: water activity; ΔE: total color change. Results were obtained in triplicate and are expressed as mean ± standard deviation.

**Table 3 foods-13-01178-t003:** Average drying rate of chayote slices during the constant drying period.

Treatment	* MR_average_	** X_c_
Fresh	0.059 ± 0.006 ^b^	5.0 ± 0.29 ^a^
Fresh-coated	0.050 ± 0.003 ^a^	5.0 ± 0.13 ^a^
Blanched	0.054 ± 0.002 ^ab^	5.6 ± 0.21 ^bc^
Blanch-coated	0.074 ± 0.01 ^c^	9.2 ± 0.86 ^d^

Mean values with different letters indicate a significant difference (*p* < 0.05) by the Fisher’s LSD test. * MR_average_ in kg water/kg of dry matter.min; ** X_c_ in kg water/kg of dry matter.

**Table 4 foods-13-01178-t004:** Values of the drying constants and coefficients from different models fitted to coated and uncoated chayote slices.

Treatment	Model			
	Page $\mathrm{MR}=\exp\left({-kt}^{n}\right)$	Henderson and Pabis MR=a exp(−kt)	Parabolic $MR=a+bt+{ct}^{2}$	Weibull $MR=exp\left(-{\left(\frac{t}{b}\right)}^{a}\right)$
Fresh	a=0.96 $k=2.52\times 10^{-4}$ n=1.57 $R^{2}=0.995$	a=1.07 k=0.006 $R^{2}=0.971$	a=1.01 b=−0.004 $c=3.92\times 10^{-6}$ $R^{2}=0.998$	a=1.46 b=188.44 $R^{2}=0.994$
Fresh-coated	a=0.96 $k=1.177\times 10^{-4}$ n=1.66 $R^{2}=0.993$	a=1.08 k=0.005 $R^{2}=0.964$	a=1.02 b=−0.003 $c=2.73\times 10^{-6}$ $R^{2}=0.997$	a=1.56 b=222.54 $R^{2}=0.992$
Blanched	a=0.97 $k=2.09\times 10^{-4}$ n=1.60 $R^{2}=0.992$	a=1.08 k=0.006 $R^{2}=0.964$	a=1.01 b=−0.003 $c=2.13\times 10^{-6}$ $R^{2}=0.999$	a=1.50 b=193.22 $R^{2}=0.991$
Blanch-coated	a=0.97 $k=3.06\times 10^{-4}$ n=1.63 $R^{2}=0.996$	a=1.09 k=0.008 $R^{2}=0.960$	a=1.03 b=−0.005 $c=6.30\times 10^{-6}$ $R^{2}=0.996$	a=1.55 b=140.67 $R^{2}=0.995$

*a*, *k*, *n*, *b*, *c* are constant of the models; R^2^: Correlation coefficient; E: Scientific notation.

**Table 5 foods-13-01178-t005:** Values of the estimated parameters and constants of the GAB, BET, and Henderson models fitted to coated and uncoated chayote slices.

Treatment	Model		
	GAB	BET	Henderson
Fresh	$M_{0}=15.37$ C=6.33 K=0.84 RSME=2.58	$M_{0}=14.14$ C=4.77 RSME=1.96	$C_{H}=6.12\times 10^{-6}$ b=1.39 RSME=2.34
Fresh-coated	$M_{0}=14.58$ C=9.98 K=0.84 RSME=3.13	$M_{0}=14.16$ C=6.54 RSME=2.84	$C_{H}=2.27\times 10^{-6}$ b=1.52 RSME=2.94
Blanched	$M_{0}=16.06$ C=9.32 K=0.78 RSME=2.51	$M_{0}=14.08$ C=7.17 RSME=1.98	$C_{H}=9.73\times 10^{-7}$ b=1.63 RSME=2.34
Blanch-coated	$M_{0}=13.14$ C=7.24 K=0.88 RSME=2.45	$M_{0}=11.29$ C=6.54 RSME=1.11	$C_{H}=4.67\times 10^{-6}$ b=1.46 RSME=1.32

*M_o_*: monolayer moisture; *C*, *K*, *C_H_*, *b*: constant of the respective model; RSME: the root mean value of the squared error of the fit of the model.

## Data Availability

The original contributions presented in the study are included in the article, further inquiries can be directed to the corresponding author.

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
