# Peer review of "Composite Coatings Applied to Fresh and Blanched Chayote (Sechium edule) and Modeling of the Drying Kinetics and Sorption Isotherms"

_foods, 2024, doi:10.3390/foods13081178_

Round 1

Reviewer 1 Report

Comments and Suggestions for Authors

Review on manuscript: foods-2928850

Composite coatings applied to fresh or blanched chayote (Sechium edule) and modeling of the drying kinetics and sorption isotherms

by Yokiushirdhilgilmara Estrada-Girón, Angelina Martín del Campo-Campos, Emmanuel Gutiérrez-García, Víctor V. A. Fernández-Escamilla, Liliana Martínez-Chávez, Teresa de Jesús Jaime-Ornelas

submitted to Foods

Research paper 

This manuscript prepared the Composite coatings applied to fresh/blanched chayote, and performed the modeling of the drying kinetics and sorption isotherms. Overall, this study was of great significance to the processing of dried chayote with higher stability and quality. However, there were still some points which need serious consideration of the authors.

Detailed recommendation:

-Abstract: this part should be re-written because of too many English errors.

-Line 115: the "Physicochemical measurements" should be changed to "Measurements of moisture, color, and rehydration capacity".

-Table 1: for a correct 3-factor and 3-level Box-Behnken design, the run numbers of the center points should be 5 rather than 3. Replenish these experiments.

-How many replicates did the authors conduct for each experiment? Please add the related information when describing the method or whitin the statistical analysis part.

-Table 2: the positions of "aw" and "ΔE" were not matched. Adjust them.

-Figure 2: what's the unit of rehydration ratio? %? Add it.

-3.2. Optimized coating: where were the resulting equations optimized by response surface analysis? As vital important results, these equations MUST be provided with statistical data, such as the coefficient estimate, sum of squares, df, mean square, F-value, and P-value for the model, linear factor, quadratic factor, interactive factor, residual, lack of fit, and pure error. Add them!

-Author Contributions: the detailed contribution of each author was not provided. Add the information.

-Funding: the grant number was missing. Add the information.

-References: the most recently published articles were very less cited. Adjust them.

Comments on the Quality of English Language

The writing English of this manuscript was not easy to read and moderate language editing is required.

Author Response

Review on manuscript: foods-2928850

Composite coatings applied to fresh or blanched chayote (Sechium edule) and modeling of the drying kinetics and sorption isotherms

by Yokiushirdhilgilmara Estrada-Girón, Angelina Martín del Campo-Campos, Emmanuel Gutiérrez-García, Víctor V. A. Fernández-Escamilla, Liliana Martínez-Chávez, Teresa de Jesús Jaime-Ornelas

submitted to Foods

Research paper 

This manuscript prepared the Composite coatings applied to fresh/blanched chayote, and performed the modeling of the drying kinetics and sorption isotherms. Overall, this study was of great significance to the processing of dried chayote with higher stability and quality. However, there were still some points which need serious consideration of the authors.

Detailed recommendation:

-Abstract: this part should be re-written because of too many English errors.

Re: Abstract was corrected and re-written in the revised manuscript.

-Line 115: the "Physicochemical measurements" should be changed to "Measurements of moisture, color, and rehydration capacity".

Re: The term was corrected in the revised manuscript as suggested.

-Table 1: for a correct 3-factor and 3-level Box-Behnken design, the run numbers of the center points should be 5 rather than 3. Replenish these experiments.

Re: The number of center points depends on the number of factors; thus, for designs of 3 factors, 3 center points are recommended, for 4 factors 5 center points and for 5 factors 7 center points. This type of design is useful because it avoids treatment combinations considered as extreme (https://online.stat.psu.edu/stat503/lesson/11/11.2/11.2.2)

-How many replicates did the authors conduct for each experiment? Please add the related information when describing the method or within the statistical analysis part.

Re: The experimental design was run by duplicate and the measurements were conducted in triplicate. This was mentioned in section 2.5 of the corrected version of the manuscript.

-Table 2: the positions of "aw" and "ΔE" were not matched. Adjust them.

Re: It was corrected in the revised manuscript.

-Figure 2: what's the unit of rehydration ratio? %? Add it.

Re: In section 2.4 it is defined that the rehydration ratio was expressed as:

      where wd is the final weight of the hydrated sample (g) and wi is the initial weight of the dried sample (g).  [Aksoy, A., Karasu, S., Akcicek, A., & Kayacan, S. (2019). Effects of different drying methods on drying kinetics, microstructure, color, and the rehydration ratio of minced meat. Foods, 8(6), 216.]

So, it is a dimensionless parameter.

-3.2. Optimized coating: where were the resulting equations optimized by response surface analysis? As vital important results, these equations MUST be provided with statistical data, such as the coefficient estimate, sum of squares, df, mean square, F-value, and P-value for the model, linear factor, quadratic factor, interactive factor, residual, lack of fit, and pure error. Add them!

Re: We provide the equations with their respective linear, quadratic, and interaction coefficients, as well as a Table with the statistical of the significant factors. Tables of the statistics are added in the appendix. Moreover, the authors reserve the optimized formulations because of a potential patent for the processing of chayote, but we share plenty of data in Table 2 and Figure 2.

-Author Contributions: the detailed contribution of each author was not provided. Add the information.

Re: It was added at the end in the revised manuscript.

-Funding: the grant number was missing. Add the information.

Re: It was added at the end in the revised manuscript.

-References: the most recently published articles were very less cited. Adjust them.

Re: It was corrected in the revised manuscript.

Comments on the Quality of English Language:

The writing English of this manuscript was not easy to read and moderate language editing is required.

Re: It was corrected in the revised manuscript.

Other changes:

We added three new references and the surface response plots of the blanched chayote.

Reviewer 2 Report

Comments and Suggestions for Authors

This manuscript is about the composite coating and their drying characteristics. Authors used Chayote for the experimentation. Manuscript written fairly well but need some changes in English to clarify the information delivered.

Following are my main concerns;

1. Title - replace 'or' with 'and' to give correct meaning to the title of the manuscript

2. Materials and Methods -

(a) Give relative humidity values of chemical salts at different temperatures, so readers are aware of the environment they need get saturated salts

(b) Blanching process should be explained fully and how you maintain the temperature at the required temperature

(c) Please add response surface equation with nomenclature and schematics for your design of the experimentation

(d) drying kinetics models needs why you use these models and how you select the best one based on what criterias should be shown

(e) never discussed about coating thickness and its effect on drying characteristics

Results and Discussion- it is not very clear what are the values given in Table 2. For example moisture after drying (why and how these values are obtained)

I cannot see any nomenclature anywhere, should need it

Comments on the Quality of English Language

Manuscript written well in English, but need to do further checking for whether sentences are written deliver intended meaning. Also check for typo Erros.

Author Response

This manuscript is about the composite coating and their drying characteristics. Authors used Chayote for the experimentation. Manuscript written fairly well but need some changes in English to clarify the information delivered.

Following are my main concerns;

  1. Title - replace 'or' with 'and' to give correct meaning to the title of the manuscript

Re: The word was corrected in the revised manuscript, as suggested.

  1. Materials and Methods -

(a) Give relative humidity values of chemical salts at different temperatures, so readers are aware of the environment they need get saturated salts

Re: The relative humidity of salts was added in section 2.8.

(b) Blanching process should be explained fully and how you maintain the temperature at the required temperature

Re: The process was described as conducted, and we also indicate that temperature was monitored with the aid of a probe tip thermometer, in section 2.2

(c) Please add response surface equation with nomenclature and schematics for your design of the experimentation

Re: The surface response equation was indicated in section 2.5, as well as the variables, factors, and coefficients of the regression. Also, in a new Table, we summarize the statistical, for the significant factor of fresh and blanche slices and added the regression equations in the results section.

(d) drying kinetics models needs why you use these models and how you select the best one based on what criterias should be shown

Re: It was added in the revised manuscript.

(e) never discussed about coating thickness and its effect on drying characteristics

Re: It was added in the revised manuscript

“About the thickness of the coatings, the optimized solutions gave films of 0.083 and 0.076 mm, for the fresh and blanched slices, respectively. Despite coatings protecting chayote from unwanted damage during the air-drying process, increasing the thickness of the coating could lead to an increase in the drying times.”

Results and Discussion:

it is not very clear what are the values given in Table 2. For example moisture after drying (why and how these values are obtained)

Re: Values in Table 2, correspond to the physicochemical measurements, while values from Table 3 were extracted from Figure 5. These data are important to mention because they represent the critical moisture content (Xc) which is the average material moisture content at which the drying rate begins to decline. The fundamentals are not included as is expected that the reader has the basic knowledge of the stages of a drying curve. Nonetheless, we mention in the text that values are obtained from Figure 5, and values at Xc are indicated in Table 3.

 “Furthermore, the critical moisture (Xc) is an important parameter that can be obtained from the drying rate curves and represents the value at which the constant rate changes to the falling rate.”

I cannot see any nomenclature anywhere, should need it

Re: The nomenclature of the compounds is mentioned in the text in the section on materials and methods, as well as the equations. However, at the foot of some tables (2, 4, and 5) we repeat the abbreviations, to spare the reader returning to the methodology section.

Comments on the Quality of English Language:

Manuscript written well in English, but need to do further checking for whether sentences are written deliver intended meaning. Also check for typo Errors.

Re: the English was corrected and revised in the whole manuscript.

Round 2

Reviewer 1 Report

Comments and Suggestions for Authors

The authors have reivsed the whole manuscript point by point based on the reviewers' comments and suggestions. Therefore, I think the revised manuscript can be completely accepted for the publication on the journal Foods.

Reviewer 2 Report

Comments and Suggestions for Authors

Thanks for answering reviewer comments and provided a revised manuscript.